

# Deep learning and support vector machines for transcription start site identification

José A. Barbero-Aparicio,  Alicia Olivares-Gil,  José F. Díez-Pastor and César García-Osorio

Departamento de Ingeniería Informática, Universidad de Burgos, Burgos, Spain

## ABSTRACT

Recognizing transcription start sites is key to gene identification. Several approaches have been employed in related problems such as detecting translation initiation sites or promoters, many of the most recent ones based on machine learning. Deep learning methods have been proven to be exceptionally effective for this task, but their use in transcription start site identification has not yet been explored in depth. Also, the very few existing works do not compare their methods to support vector machines (SVMs), the most established technique in this area of study, nor provide the curated dataset used in the study. The reduced amount of published papers in this specific problem could be explained by this lack of datasets. Given that both support vector machines and deep neural networks have been applied in related problems with remarkable results, we compared their performance in transcription start site predictions, concluding that SVMs are computationally much slower, and deep learning methods, specially long short-term memory neural networks (LSTMs), are best suited to work with sequences than SVMs. For such a purpose, we used the reference human genome GRCh38. Additionally, we studied two different aspects related to data processing: the proper way to generate training examples and the imbalanced nature of the data. Furthermore, the generalization performance of the models studied was also tested using the mouse genome, where the LSTM neural network stood out from the rest of the algorithms. To sum up, this article provides an analysis of the best architecture choices in transcription start site identification, as well as a method to generate transcription start site datasets including negative instances on any species available in Ensembl. We found that deep learning methods are better suited than SVMs to solve this problem, being more efficient and better adapted to long sequences and large amounts of data. We also create a transcription start site (TSS) dataset large enough to be used in deep learning experiments.

# INTRODUCTION

Since the emergence of next-generation sequencing (NGS) methods, understanding gene expression has become one of the most important goals in bioinformatics. Therefore, the

Corresponding author
José A. Barbero-Aparicio,
jabarbero@ubu.es

identification of each component affecting transcription is a key task in this field. Promoters stand out among these components, being closely related to transcription initiation and inhibition. Inside the promoter, a particular position is especially relevant: the transcription start site (TSS). Being able to locate the TSS means locating the core promoter, which can give us important information about where a particular transcript is initiated. Since the use of the terms TSS and TIS in the literature is sometimes confusing, we want to clarify that here we understand TSS as the base in DNA where the transcription of mRNA begins, and the translation initiation site (TIS) as the codon of the mRNA where the protein synthesis begins.

For TIS predictions, there is increasing evidence in recent publications that this problem is practically solved (*Wei et al., 2021*), but TSS is a more difficult problem that poses several additional difficulties. This is mainly due to the possibility that more than one TSS exists in the same promoter, but also due to the lack of available sequences, since some databases only include sequences of interest downstream of the TSS.

One of the consequences of better and cheaper sequencing technologies is the wide availability of sequencing data, access to which has been made easier by international efforts to have centralized databases accessible to researchers. *Kodama, Shumway & RL (2011)* provide a good list of the main databases and present an interesting analysis of the main contributors, sequencing technologies, sequenced organisms, types of studies, data file format and metadata models.

This accessibility to large volumes of raw sequencing data explains the emergence of numerous papers in which machine learning techniques are applied to the identification of key elements in DNA sequences. *Kim et al. (2020)* use a deep learning model, DeepTFactor, with three parallel subnets, each with three convolutional layers, to detect transcription factors in amino acid sequences, obtaining performance higher than TFpredict (*Eichner et al., 2013*), an SVM model.

*Osmala & Lähdesmäki (2019)* present a new method called the PRobabilistic Enhancer Prediction Tool (PREPRINT) to predict genomic regulatory enhancers using chromatin feature data. PREPRINT consists of several data processing and analysis steps, including the statistical modeling of the coverage of the reference genome with the chromatin reads as a Poisson distribution, whose mean parameter is estimated with two approximations, using maximum likelihood (PREPRINT ML) and using Bayesian estimation (PREPRINT Bayesian). From this, two probabilistic distance measures are used to obtain a matrix of scores that is used to train an SVM classifier with a Gaussian kernel.

ADAPT-CAGE (*Georgakilas, Perdikopanis & Hatzigeorgiou, 2020*) is a tool for TSS prediction that uses CAGE reads, Polymerase II motifs, and various DNA features (duplex disrupt energy, duplex free energy, bending stiffness, denaturation, stacking energy, bendability, propeller twist, z-DNA, A-philicity, nucleosome positioning, protein deformation, B-DNA twist and protein-DNA twist). ADAPT-CAGE is a sophisticated combination of models that includes several SVMs and Stochastic Gradient Boosting trained with the different individual characteristics and whose outputs are combined to obtain the final prediction.

In the context of gene promoter identification, *Xu, Zhang & Lu (2011)* combine the use of three statistical divergences (Kullback–Leibler, Symmetric and Jensen–Shannon), to choose the best *k*-mers as differentiating features, three sparse autoencoder, and three SVMs with radial basis function kernels to identify promoters in the human genome. Also, for this problem, *Bhandari et al. (2021)* compare the performance of two deep learning methods (one based on CNN and another using LSTM) and a machine learning method (an ensemble of random trees). They use three different eukaryotic genomes, Yeast, A. Thaliana and Human. In addition to one-hot encoding, they apply frequency-based tokenization (FBT) for different *k*-mer sizes. Their conclusion is that the best results are achieved with FBT and the CNN-based architecture.

A common denominator of all of these works is that, in the experimental validation of the proposed methods, an important effort has been made to obtain a dataset adequate for the application of the machine learning method. This involves steps such as curation of sequences retrieved from public databases, application of data preprocessing methods, and generation of negative examples, among others.

Much progress has been made since the first methods based on the recognition of certain subsequences, motifs, or sequence features (*Bajic et al., 2006*) were developed, being deeply influenced by the expertise of researchers, *e.g.*, TATA boxes, CCAAT boxes, GpG islands, *k*-mer frequencies, GC content, *etc.* (*Werner, 1999*; *Abeel, de Peer & Saeys, 2009*; *Tatarinova et al., 2013*; *Jorjani & Zavolan, 2013*; *Shahmuradov et al., 2016*; *Bajic et al., 2004*).

However, the results obtained by these approaches did not agree with the promoter frequencies in the human genome estimated by experts (*Pedersen et al., 1999*). The success in other areas of multilayer perceptrons (*Mahdi & Rouchka, 2009*), support vector machines (*Sonnenburg, Zien & Rätsch, 2006*) and convolutional neural networks (*Pachganov et al., 2019*) has motivated their use in the context of bioinformatics as well.

While in TIS prediction several methods based on deep neural networks (DNNs) have emerged (*Wei et al., 2021*; *Zhang et al., 2017*; *Zuallaert et al., 2018*), in TSS prediction, SVMs are yet the most popular models (*Schaefer et al., 2010*; *Sonnenburg, Zien & Rätsch, 2006*; *Ohler, 2006*; *Towsey, Gordon & Hogan, 2006*) even in recent times (*Georgakilas, Perdikopanis & Hatzigeorgiou, 2020*). Due to the predominance of SVMs, it is hard to find deep learning approaches to TSS identification, and the few available studies that do use deep learning for TSS identification (*Mahdi & Rouchka, 2009*; *Zheng, Li & Hu, 2020*) do not take into account SVMs as the reference model in this area. Deep learning methods have been proven effective in related tasks mentioned before, but TSS identification presents certain aspects that can make this problem harder to solve than TIS identification, due to the different nature of the sequences and the patterns usually found in them. This kind of problems can be solved by using more complex models, as long as there is enough data to train them. In this work, we prove that, even though SVMs are well established in the transcription start site prediction field, because of their simplicity and explainability, deep learning methods can be vastly superior in their results. To test that hypothesis, we have developed a way to generate a dataset from the GRCh38 version of the human genome, since, to the best of our knowledge, there is no publicly curated and ready-to-use dataset

to obtain machine learning models to solve this specific problem. The availability of this data set will facilitate the work of other researchers in this field. Furthermore, its size, with more than a million instances, is suitable for obtaining models using deep learning techniques. This dataset and the steps to replicate the creation of the dataset are publicly available in GitHub (https://github.com/JoseBarbero/EnsemblTSSPrediction) allowing future researchers to create a similar dataset with any other species available in Ensembl. Additionally, we include a study of the proper positive to negative instances ratio in the dataset, given that the negative instances are generated in a semisynthetic way. Also, we analyze the performance of several deep neural network architectures, since very different options have been proven to be effective in related problems, although there is no standard architecture established as the state-of-the-art for TSS prediction. Finally, we tested the prediction performance of deep learning and SVM models using a similar dataset based on the mouse genome in order to study the generalization capabilities of each algorithm.

## METHODS

### Generation of the dataset

We used the reference sequence of the human genome in its *GRCh38.p13* version to have a reliable data source for our experiments. We chose this version because it is the most recent one available in Ensembl at the moment. However, the DNA sequence by itself is not enough, the specific TSS position of each transcript is needed. In this section, we explain the steps followed to generate the final dataset (Fig. 1). These steps are: raw data gathering, positive instances processing, negative instances generation, and data splitting by chromosomes.

First, we need an interface to download the raw data, which are made up of every transcript sequence in the human genome. We use *Ensembl release 104* (*Howe et al., 2020*) and its utility *BioMart* (*Smedley et al., 2009*), which allows us to easily obtain large amounts of data. It also enables us to select a wide variety of interesting fields, including the transcription start and end sites. After filtering instances that present null values in any relevant field, this combination of the sequence and its flanks will form our raw dataset.

Once the sequences are available, we find the TSS position (given by Ensembl) and the two following bases to treat it as a codon. After that, 700 bases before this codon and 300 bases after it were concatenated, giving the final sequence of 1,003 nucleotides that is going to be used in our models. These specific window values have been used in *Bhandari et al. (2021)* and we have kept them because we find them interesting for comparison purposes.

One of the most sensitive parts of this dataset is the generation of negative instances. We cannot obtain this kind of data directly, so we need to generate them synthetically. To get examples of negative instances, *i.e.,* sequences that do not represent a transcript start site, we select random DNA positions inside the transcripts that do not correspond to a TSS. Once we have selected the specific position, we obtain 700 bases ahead and 300 bases after it, as we did with the positive instances. Regarding the positive to negative ratio, in a similar problem, but studying TIS instead of TSS (*Zhang et al., 2017*), a ratio of 10 negative instances to each positive one was found optimal. Following this idea, we select 10 random

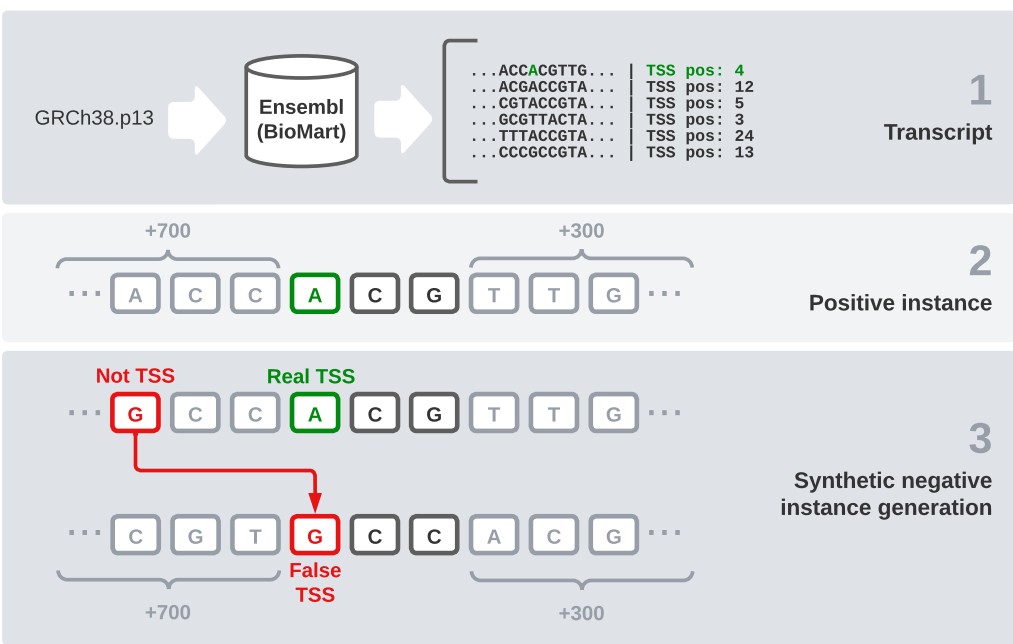

**Figure 1** Dataset generation process.

positions from the transcript sequence of each positive codon and label them as negative instances.

After this process, we end up with 1,122,113 instances: 102,488 positive and 1,019,625 negative sequences. To validate and test our models, we need to split this dataset into three parts: training, validation, and testing. We have decided to make this differentiation by chromosomes, as in *Perez-Rodriguez, Haro-Garcia & Garcia-Pedrajas (2020)*. Thus, we use chromosome 16 as validation because it is a good example of a chromosome with average characteristics. Then we selected samples from chromosomes 1, 3, 13, 19 and 21 to be part of the test set and used the rest of them to train our models.

Every step of this process can be replicated using the scripts available in https://github.com/JoseBarbero/EnsemblTSSPrediction.

## Sequence encoding

Most machine learning methods cannot read DNA sequences directly. One of the most popular ways to encode nucleotides strings is one-hot encoding. Using this method, we represent every base using a binary array of four positions, each representing one possible base. Only one of the values of the array will be 1, while the others remain 0. Hence, we have a simple way to represent four different characters: A, C, G, and T. Although some approaches are using word embedding systems based on the natural language processing field (*Wu et al., 2021*), we found one-hot encoding adequate for the purposes of this work because of its simplicity. This is the encoding used for the deep learning methods. For SVM methods, we have used both this encoding (for SVM with RBF kernel) and directly the raw sequence (for SVM with string kernel).

## Support vector machines (SVMs)

SVMs (*Cortes & Vapnik, 1995*) are one of the most used methods in machine learning and specially in binary classification. We find that SVMs are one of the few alternatives that can keep up with DNNs in complex problems such as TSS identification. These methods have some advantages over DNNs, like the possibility of introducing "biases" based on human expertise in our models through its kernel. Since this is essentially a string classification problem, we need to use a string kernel with our SVM. In *Perez-Rodriguez, Haro-Garcia & Garcia-Pedrajas (2020)*, it was found that the *weighted degrees* kernel consistently gives the best results in the identification of TIS. Due to the similarity of the problem we are trying to solve, we decided to use a weighted degree kernel as well. This kernel is also the one that gave the best results in *Sonnenburg et al. (2007)*, where they were compared with Markov chains and SVMs using other kernels.

SVM algorithms integrate a mathematical function named kernel. The purpose of this kernel is to implement the trick used by the SVM algorithms that allows instances to be represented as a relationship metric between two data points instead of being explicitly represented in a higher-dimensional space. In the case of string kernels, it is specially useful to get numerical values from the string sequences, which originally could not be used by the models.

As mentioned before, we used the *weighted degrees* (WD) kernel. This type of kernel allows us to represent the similarity of two sequences by counting the amount of $k$-mers of size $k \in \{1, \ldots, d\}$ that two specific sequences $\mathbf{s}_i$ and $\mathbf{s}_j$ of length $L$ share. This function is defined by the Eq. (1), where $\mathbb{I}$ returns 1 when the $k$-mer starting in position $l$ is the same in sequences $\mathbf{s}_i$ and $\mathbf{s}_j$ and 0 if not. $\beta_k$ is a weight factor defined by $\beta_k = 2 \frac{(d-k+1)}{d(d+1)}$.

$$K(\mathbf{s}_i, \mathbf{s}_j) = \sum_{k=1}^{d} \beta_k \sum_{l=1}^{L-k+1} \mathbb{I}\big(\mathbf{u}_{k,l}(\mathbf{s}_i) = \mathbf{u}_{k,l}(\mathbf{s}_j)\big) \tag{1}$$

To have a similar model to evaluate the benefits of a string kernel SVM, we additionally used a numerical method, specifically an SVM with a radial basis function (RBF) kernel. This kernel is commonly used in machine learning with SVM classifiers. Experimenting with this kernel allows us to see the differences in performance between a string kernel and a numerical one. In this case, the kernel function is described by Eq. (2), which is defined by the squared Euclidean distance of two instances $\mathbf{x}$ and $\mathbf{x}'$ and a $\gamma$ value that determines the influence of the support vectors in the model.

$$K(\mathbf{x}, \mathbf{x}') = \exp\big(-\gamma \parallel \mathbf{x} - \mathbf{x}' \parallel^2\big). \tag{2}$$

The use of the RBF kernel requires the one-hot method to encode the instances, given that this is a numerical kernel and it is not possible to use the string instances as we did with the weighted degrees kernel.

All the parameters in Eqs. (1) and (2) as well as the regularization parameter in the support vector classifier model were defined by an extensive grid search cross-validation process.

Another advantage of SVMs is their interpretability, as opposed to the black-box nature of DNNs. Given the robust mathematical theory behind the method and the simplicity

that characterizes the kernels, the models are much easier to understand for humans than DNNs. This simplicity also makes the model easier to tune, having fewer parameters to optimize, needing less memory to store the trained model and having a lighter and faster predictive model (*Yoon et al., 2011*). It is also a method without the problems of local minima, being, in principle, less prone to overfitting.

Several works can be found on SVMs applied to DNA sequences that demonstrate a good performance on similar problems. Specifically, great results were obtained in alternative splicing recognition (*Rätsch, Sonnenburg & Schölkopf, 2005*), promoter (*Wang et al., 2018*; *Sato, 2018*; *Cassiano & Silva-Rocha, 2020*) and promoter features prediction (*Meng et al., 2017*), transcription start identification in eukaryotes (*Sonnenburg, Zien & Rätsch, 2006*; *Georgakilas, Perdikopanis & Hatzigeorgiou, 2020*) and prokaryotes (*Towsey, Gordon & Hogan, 2006*) or translation initiation sites prediction (*Perez-Rodriguez, Haro-Garcia & Garcia-Pedrajas, 2020*). SVMs have also been combined with convolutional neural networks (*Qian et al., 2018*). This approach will be addressed in the next section.

The string kernel has been applied using Scikit-learn's (*Pedregosa et al., 2011*) support vector classifier implementation, which allows using custom kernels. However, the kernel needs to be precomputed. This implies creating the *gram matrix* for the train and test datasets, being a computationally slow process. We tried to minimize this issue by parallelizing the process, although it is still noticeably slower than the deep learning methods that will be presented in the next section. The main parameter to choose is the value *d*, which represents the maximum length of *k*-mers to consider. We conducted a preliminary experiment using a reduced dataset, formed by a sample of 1% of the positive and negative instances, in order to test several values of *d*, the results of which can be seen in Table 1. Higher values generally achieve better results, but at a much higher running time. As we will address later, training times can be significant having to calculate the kernel gram matrix: a problem implying quadratic growth. For that reason, we determined that a *d* value of 10 is reasonable in this case. It improves the results significantly, being a bit less precise than 20 but at a considerably lower running time.

The most relevant inconvenience of support vectors machines are their memory and execution time requirements when used with very large data sets, precisely the data sets that appear in bioinformatics. For reference, in one of our preliminary experiments we needed to use a server with 1 TB of RAM, and the experiment took several months to complete, even parallelizing using 20 cores. These execution times were reduced in subsequent experiments using an approach similar to that suggested in *Graf et al. (2004)*, the Cascade SVM. Specifically, we have used the implementation of the method available in https://github.com/fhebert/CascadeSVC for the RBF kernel, and a modified version to work with the WD kernel. The idea of this method is to get the support vectors by levels. In the first level, the original dataset is divided into several subsets (typically a number power of two), then an SVM is used on each subset to obtain a set of support vectors. In the next level of the cascade, two sets of support vectors from the previous level are combined and used on a new SVM, which in turn will give a set of support vectors that will be combined with others at the next level of the cascade. In the last level of the cascade, with only one SVM, the final set of support vectors that is obtained is added to each of the subsets of the

**Table 1** SVM with weighted degree kernel results with different d values.

| d value | Acc | BC | F1 | AUC ROC | Time (s) |
|---|---|---|---|---|---|
| 1 | 0.82762 | 5.9535 | 0.2179 | 0.6460 | 202.63 |
| 3 | 0.8982 | 3.5130 | **0.2747** | 0.7292 | 522.51 |
| 10 | **0.9105** | **3.0880** | 0.1216 | 0.7834 | 917.69 |
| 20 | 0.9095 | 3.1224 | 0.0269 | **0.7971** | 1369.52 |
| 50 | 0.9080 | 3.1771 | 0.0008 | 0.7897 | 2567.18 |

**Notes.**
Bold values represent the best results.
Acc, Accuracy; BC, Binary Cross-Entropy; F1, F1-metric; AUC ROC, Area Under the Receiver Operating Characteristic Curve.

first level of the cascade, and the process is repeated for a few additional iterations until convergence is reached. The application of this procedure allows reducing the duration of an experiment from several months to several hours. It also allows reducing the memory requirements.

## Deep learning

It could be said that the emergence of deep learning was a turning point for artificial intelligence and, recently, also in bioinformatics. DNNs have set a new state-of-the-art performance in tasks like variant calling (*Poplin et al., 2018*) or protein folding prediction (*Jumper et al., 2021*), and have been found very valuable in promoter identification (*Bhandari et al., 2021*). Since the first work using neural networks in promoter prediction (*Demeler & Zhou, 1991*), models such as multilayer perceptrons (MLP) (*Shahmuradov, Umarov & Solovyev, 2017*), convolutional neural networks (*Umarov et al., 2019*; *Pachganov et al., 2019*; *Wei et al., 2021*) or a combination of CNN and recurrent neural networks (RNNs) (*Oubounyt et al., 2019*; *Zhang et al., 2017*) have appeared in problems related to promoter identification. Meanwhile, it is much harder to find deep learning approaches for TSS identification than other related problems, and the ones available (*Mahdi & Rouchka, 2009*; *Zheng, Li & Hu, 2020*), do not offer a comparison that takes into account SVMs as the most established method in this field.

Unlike SVMs, DNNs' interpretability is a challenging task, but on the other hand, they do not require as much expertise in the field of application as with SVMs kernel design. DNNs can learn the particularities of the problem by themselves. Even though DNNs do not need as much knowledge in the molecular biology field as SVMs do to create appropriate kernels, they do need that expertise in the deep learning field to be able to develop and fine-tune the model. Furthermore, DNNs need significantly more data to be trained in complex problems such as TSS prediction, and can fall in local minima and tend to overfit more than SVMs (*Salman & Liu, 2019*).

In *Khan et al. (2020)*, in the context of promoter identification, the researchers concluded that MLPs can achieve better results than SVMs. We believe that extending this research to TSS identification could be promising. Furthermore, string kernels were not taken into account, only RBF and sigmoid kernels were tested, and we think that the inclusion of CNNs and RNNs will enrich the study.

We selected three of the most representative types of DNNs that can work with one-hot encoded sequences: a CNN, a LSTM (*Hochreiter & Schmidhuber, 1997*) and a combination of both. We chose these architectures because they are well established in the deep learning field and are good examples of the kind of models we can find used in related problems research (*Wei et al., 2021*; *Zhang et al., 2017*; *Zuallaert et al., 2018*).

In the experiments, the deep learning models have been optimized using the validation dataset mentioned in Section 2.1. The values of the hyperparameters have been found using a grid search cross-validation process. A batch size of 32, a learning rate of 0.001, and the Adam optimizer (*Kingma & Ba, 2017*) were used in the experiments for every deep learning model. The number of epochs was set to 100, although an early stopping mechanism would stop the training process after 10 epochs without any improvement. All DNNs have been trained by applying binary cross entropy as their loss function.

Later, in the experiments, the architectures used for the neural networks are faithful to those used in the papers in which these architectures are presented, with a number of parameters ranging from 100,000 to 600,000. Modifications that increase the number of parameters have been avoided, because this could lead to generalization problems in some of the models.

### Convolutional neural networks (CNNs)

CNNs are a kind of artificial neural network based on the use of filtering operations called convolutions, which are usually used as a complement to classic perceptrons and pooling layers. These convolutions allow the network to represent the input information with different feature maps. CNNs have excelled in tasks related to image classification (*Krizhevsky, Sutskever & Hinton, 2012*), to the point that it has even led scientists to look for ways to encode all kinds of data as if they were images (*Sharma et al., 2019*) and turn different problems, for example, variant calling (*Poplin et al., 2018*), into image classification tasks, in which CNNs can achieve excellent results (*Nguyen et al., 2016*). In this case, we are working with a similar representation of the data, coming from DNA sequences of 1003 bases one-hot encoded in arrays of length 4. Thus, having $N$ samples, we start with an input of $N \times 1003 \times 4$, which can be seen as a set of 1D images with four channels.

The structure of the CNN (Fig. 2A) is based on *Bhandari et al. (2021)* considering that it has been proven effective and that we want to keep similar structures with comparison purposes. Therefore, the network is composed by three 1D convolution layers of size 5, stride 1 and 32 filters. Each convolution layer is followed by a max-pooling layer with size 4. Following these steps, there are three fully connected layers with 1,024, 512 and 128 units, each one followed by a 20% dropout step (*Salman & Liu, 2019*). Finally, there is a classification layer using the sigmoid activation function.

### Long short-term memory networks (LSTMs)

Recurrent neural networks (RNN) are a type of artificial neural network designed to work with sequential problems. LSTMs are a specific kind of RNNs developed to improve RNNs efficiency while solving issues like the vanishing gradient problem (*Hochreiter, 1998*).
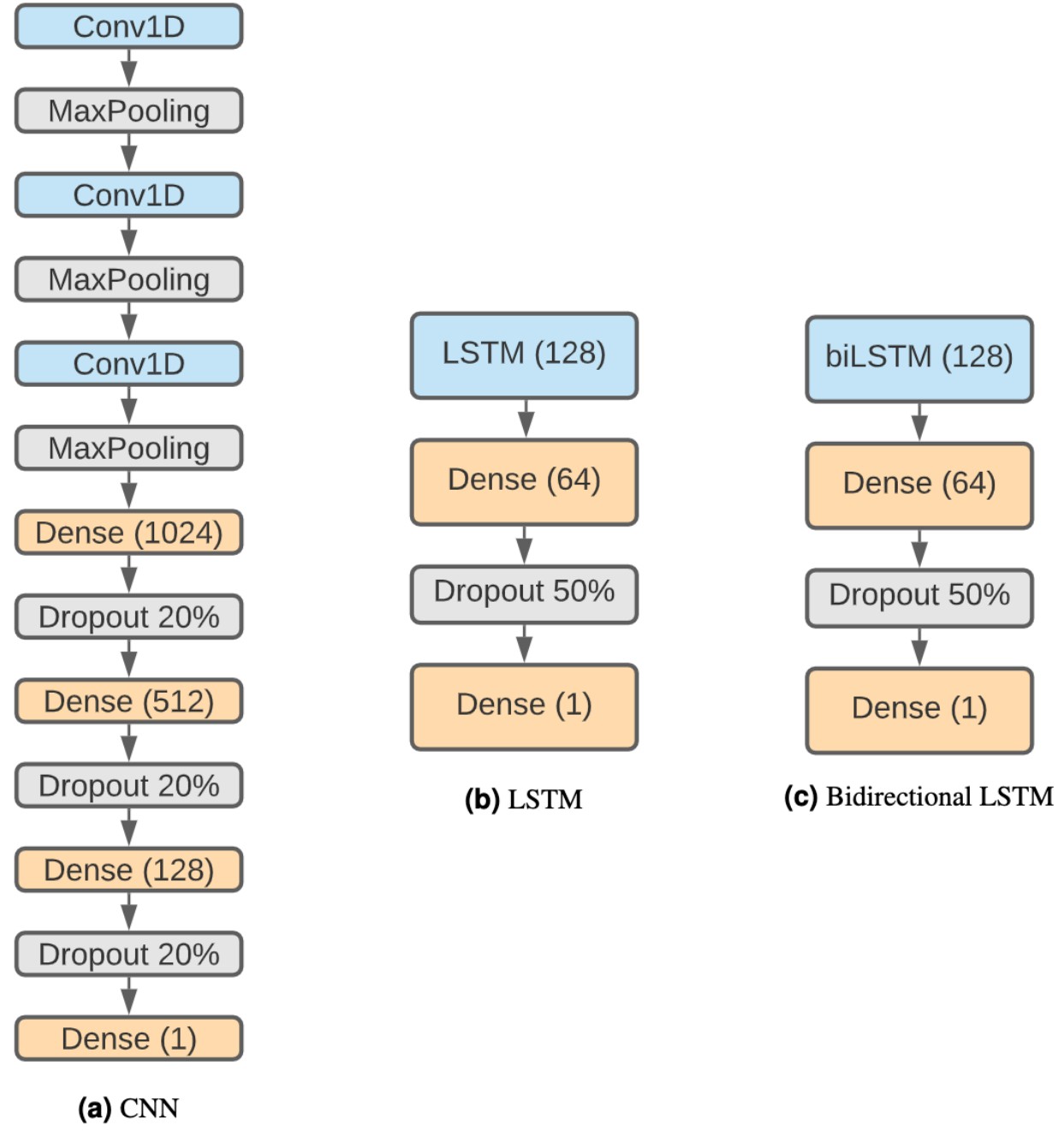

**Figure 2** **DNN architectures.** (A) CNN. (B) LSTM. (C) Bidirectional LSTM.

LSTMs (*Hochreiter & Schmidhuber, 1997*) are one of the best-suited structures for our problem, as they are designed to work with sequences, and DNA is a perfect example. The decision to use this neural network architecture is influenced by the results obtained with it in *Bhandari et al. (2021)*, but in addition to the regular LSTM, a bidirectional LSTM has also been used, as both have been shown to be effective in related problems such as DNA

protein binding identification (*Zhang et al., 2019*). The disadvantages of LSTMs are their higher computational cost and that their parallelization is harder than CNNs, making this approach much slower.

The networks are composed of a regular (see Fig. 2B) or a bidirectional LSTM (Fig. 2C) layer of 128 units followed by a dense layer of 64 units and a 50% dropout step. Finally, the last layer is a binary classification using a sigmoid function.

### *LSTM + CNN*

Finally, we have decided to use a model formed by a combination of CNNs and LSTMs. This kind of model has been shown to be effective in *Zhang et al. (2017)* and *Zhang et al. (2019)*, achieving remarkable results in TIS and prediction of DNA proteins. The higher complexity of this method makes it capable of finding more elaborate patterns. Our architecture (Fig. 3A) starts with a 1D convolution layer of 64 filter and size 3, followed by a max pooling layer of 3 and a 25% dropout. After that, there is an LSTM layer of 64 units and an 80% dropout to avoid overfitting. The output classification is the result of a fully connected layer with sigmoid activation.

We also consider a different version of this model using a bidirectional LSTM (Fig. 3B) instead of a regular LSTM to assess whether the results compensate for the additional complexity of the model.

## RESULTS

### Metrics

To evaluate the models, we have employed different metrics, each adapted to a different context. The first one is the accuracy (*Acc*) (Eq. (3)) and it is acceptable as a first approach to measure model performance.

$$Acc = \frac{Number\ of\ correct\ predictions}{Total\ number\ of\ predictions\ made}. \tag{3}$$

However, accuracy is not the best option in situations involving imbalanced data. Because of that, we decided to use more appropriate metrics, like binary cross-entropy (*BC*) (Eq. (4)) which is a standard metric in binary classification

$$BC = -\frac{1}{N}\sum y_i \cdot \log(p(y_i)) + (1 - y_i) \cdot \log(1 - p(y_i)) \tag{4}$$

where $y$ is the actual class and $p(y_i)$ is the probability that the model assigns to the instance $i$ being positive. It will be 1 if the instance belongs to the positive class and 0 otherwise. A good model gives low values of *BC*.

Another metric that works well in the context of imbalanced data is the F1-score (Eq. (7)) which takes the harmonic mean of the precision and recall metrics, that are defined in terms of the true positives (TP), false positives (FP), true negatives (TN) and false negatives (FN) all obtained from the confusion matrix.

$$Precision = \frac{TP}{TP + FP} \tag{5}$$

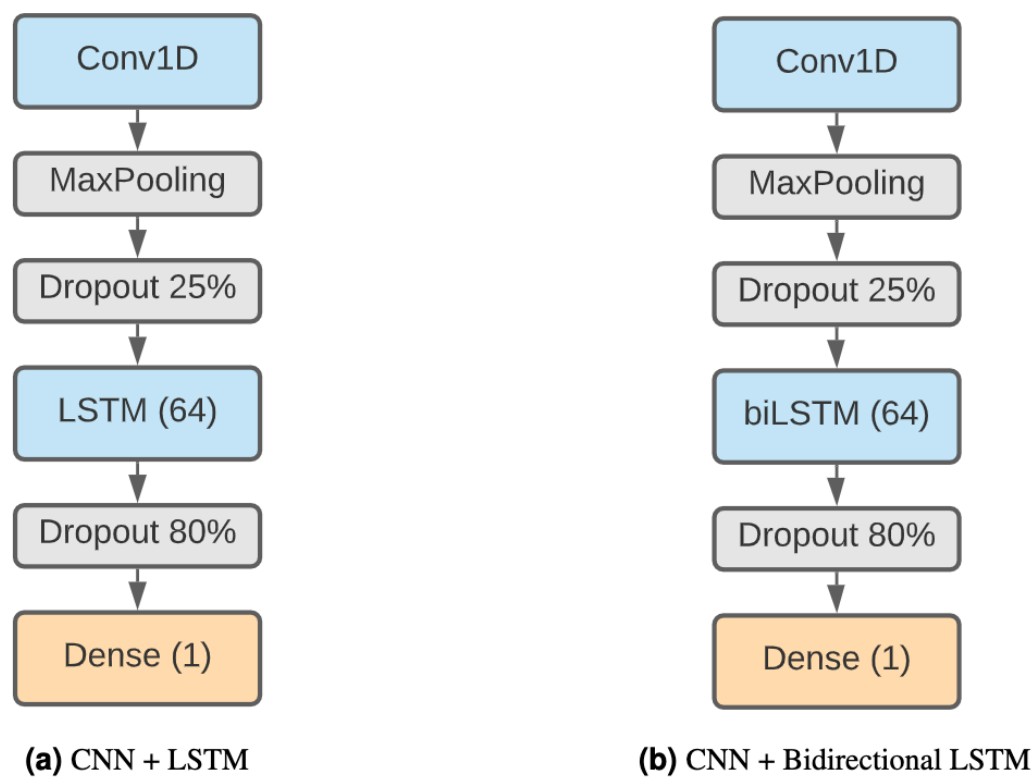

**(a)** CNN + LSTM          **(b)** CNN + Bidirectional LSTM

**Figure 3** **Combined DNN architectures.** (A) CNN + LSTM. (B) CNN + bidirectional LSTM.

$$Recall = \frac{TP}{TP + FN} \tag{6}$$

$$F1 = \frac{2 \cdot Precision \cdot Recall}{Precision + Recall}. \tag{7}$$

Finally, we used the area under the ROC curve (AUC), a thorough metric for imbalanced datasets, which is the area under the curve when plotting the sensitivity ($Sn$) against the specificity ($Sp$):

$$Sn = \frac{TP}{TP + FN} \tag{8}$$

$$Sp = \frac{TN}{FP + TN}. \tag{9}$$

The AUC values are between 0 and 1 with values closer to 1 associated with better predictions.

Each metric in this study was calculated using a five-fold validation process. This process has a minor peculiarity since the training, validation and test sets are defined by

chromosomes, which prevents the use of a classical five-fold cross-validation. In the case of the deep leaning models, we retrained the models five times using different seeds. In the cascade SVM experiments, we used random splits of the dataset in the first step of the cascade process, which leads to five independent results.

## Evaluation of the positive to negative ratio in the dataset generation

In this article, we also wanted to test whether the use of a ratio of one to ten between positive and negative instances is the correct choice, as suggested in *Zhang et al. (2017)*. Different executions have been run with each method using a one to one and a one to ten ratio between positive and negative instances. Although for the training process a ratio of one to one has been used, for the test process, a one to ten ratio has been kept, given this is the approximated ratio we could find in real data.

It consists of generating only one negative instance for each positive one instead of ten, which could be considered a process of undersampling on the majority class.

We evaluated two methods to obtain a one to one ratio of positive and negative instances. The first is very simple, since the negative instances are generated synthetically with the dataset. It consists of generating only one negative instance for each positive one, instead of ten, which could be considered a process of undersampling on the majority class. For the second method, we used SMOTE (Synthetic Minority Oversampling Technique), which is based on generating synthetic instances to oversample the minority class, leading to a ratio of one positive to one negative instance. The downside of SMOTE is that it needs to be applied to numerical values, and the weighted degree kernels work with sequences. This means that SMOTE can not be applied to our string kernel method.

As we can see in Table 2, most of the methods get the best results when they are trained with a positive to negative ratio of one to ten, instead of one to one. Also, using SMOTE does not seem to improve the performance either. This seems to be in line with the results reported in *Blagus & Lusa (2013)* where the authors claim that SMOTE does not work well with large numbers of features. This is precisely what is happening here, where in order to apply SMOTE we are using one-hot encoding, what increases to 4012 the number of characteristics of each sequence.

The conclusion that can be drawn from these results is that trying to balance the data set, for example, applying a subsampling to the majority class, is not a good heuristic to improve the results in this type of problem. Nor does any improvement seem to be obtained by generating artificial instances with SMOTE. For this reason, the rest of the experiments will be executed applying a positive to negative ratio of one to ten. Finally, from these results we can deduce that the use of more negative instances is helping to obtain better models and that the method chosen to generate these instances is adequate.

## Comparing SVM and deep learning results

The first magnitude that we take into account is the training time of each model. Given that the model only has to be trained once, it could be reasonable to use a model whose training is slower if the predictions are significantly better. In this case, we find notable

**Table 2  Results training with 10 negative instances for each positive one and one negative instance for each positive one.**

| Classifier | Acc | BC | F1 | AUC ROC | Ratio |
|---|---|---|---|---|---|
| CNN | 0.9176 | **0.2199** | **0.4968** | **0.8682** | |
| LSTM | **0.9354** | **0.1776** | **0.5771** | **0.9144** | |
| BLSTM | **0.9383** | **0.1697** | **0.6105** | **0.9216** | |
| CNN+LSTM | **0.9329** | **0.1806** | **0.5913** | **0.9121** | 1 to 10 |
| CNN+BLSTM | **0.9325** | **0.1817** | **0.5880** | **0.9106** | |
| SVM WD | **0.9090** | 3.1433 | 0.4124 | 0.8232 | |
| SVM RBF | 0.5044 × | 17.1162 | 0.2386 | **0.7737** | |
| CNN | 0.8946 × | 0.3487 × | 0.4831 | 0.8629 | |
| LSTM | 0.9108 | 0.4118 × | 0.5316 | 0.8828 | |
| BLSTM | 0.9116 | 0.3878 × | 0.5420 | 0.8914 | |
| CNN+LSTM | 0.9078 | 0.3535 × | 0.5407 | 0.8948 | 1 to 1 |
| CNN+BLSTM | 0.8983 | 0.3646 × | 0.5033 | 0.8686 | |
| SVM WD | 0.7757 | 7.7438 × | 0.3755 | **0.8429** | |
| SVM RBF | **0.6121** | **13.3971** | **0.2744** | 0.7614 | |
| CNN | **0.9355** | 0.2594 | 0.4341 × | 0.7773 × | |
| LSTM | 0.8885 × | 0.2497 | 0.4590 × | 0.8108 × | |
| BLSTM | 0.8993 × | 0.2496 | 0.4644 × | 0.8113 × | |
| CNN+LSTM | 0.8910 × | 0.2444 | 0.4677 × | 0.8221 × | SMOTE |
| CNN+BLSTM | 0.8935 × | 0.2469 | 0.4642 × | 0.8215 × | |
| SVM WD | — | — | — | — | |
| SVM RBF | 0.5919 | 14.0944 × | 0.2608 × | 0.7587 × | |

**Notes.**
The best values are highlighted in bold, and the worst values marked with ×
Acc, Accuracy; BC, Binary Cross-Entropy; F1, F1-metric; AUC ROC, Area Under the Receiver Operating Characteristic Curve.

differences between the methods in Fig. 4, where the training times for each model can be seen.

The times for the deep learning methods vary from nearly 2 h of the CNN to 12 h of the bidirectional LSTM, while for the Cascade SVM method we got execution times of almost 4 days using the string kernel. On the other hand, the SVM with an RBF kernel can be trained in 2 h using the cascade approach and the Intel(R) Extension (https://github.com/intel/scikit-learn-intelex) for Scikit-learn (which, unfortunately, cannot be used when the SVM uses a string kernel). Without the cascade approach, the SVM with a string kernel could take several months before being able to complete the training process.

This has to do with the weighted degree kernel. This kind of kernel can achieve notably better results working with strings, as seen in *Perez-Rodriguez, Haro-Garcia & Garcia-Pedrajas (2020)*, but it comes at a great computational cost. In this case, the kernel has been developed in Python using the Scikit-learn (*Pedregosa et al., 2011*) tool to use personalized kernels, which implies precomputing the entire kernel matrix. This kernel computation can be optimized using faster languages like C, so we implemented the gram matrix calculation in C to reduce computational time. To minimize these problems, the matrix computing

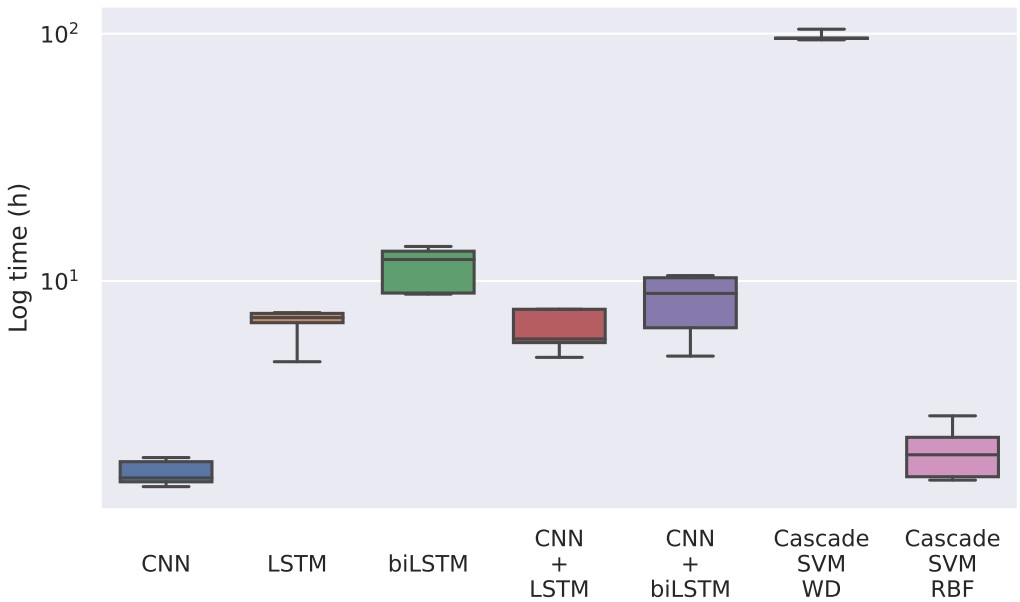

**Figure 4** Training time for each model in logarithmic scale.

step has also been programmed to be parallelized. In our case, the SVM experiments have been run on an Intel(R) Xeon(R) Gold 6150 CPU @ 2.70 GHz using 20 cores. Despite that, its training process is remarkably slower than that of the other methods.

In order to study the effects of the amount of training data on the performance of each method we performed experiments using the 10% (Table 3) and 50% (Table 4) of the data in order to have a fair comparison between SVM and the rest of the algorithms, given that SVM methods could perform better than deep neural networks in situations with fewer data available. Now, the reduction in training times achieved by working with smaller datasets has allowed us to include regular SVM methods to analyze the effects of cascading optimization on the results.

In Table 3 we can see how, even using only 10% of the data, SVM gets the worst results due to a strong overfitting (note that the regularization parameter was the best found after testing several values using a grid search), except in the case of the RBF kernel using cascade SVM, which makes the model generalize better but notably reduces its prediction performance. These results seem to indicate that, although SVMs are typically more robust to overfitting than deep networks, for this particular problem the measures against overfitting of deep networks, the dropout layers, are more effective than the regularization of SVM. When using half of the data, in Table 4 we can see how, even though the results experience a slight improvement, the SVM models keep overfitting with significantly lower results than the rest of the methods. As we can see, SVMs consistently gets the worst results in every metric but accuracy, which is related to the unbalanced nature of the data. In metrics more suited to unbalanced data, SVM performs significantly worse than the deep learning methods.

**Table 3** Results using 10% of the data.

| Classifier | Train | | | | Test | | | |
|---|---|---|---|---|---|---|---|---|
| | Acc | BC | F1 | AUC ROC | Acc | BC | F1 | AUC ROC |
| CNN | 0.9171 | 0.2154 | 0.2319 | 0.8710 | 0.9124 | 0.2360 | 0.1811 | 0.8269 |
| LSTM | 0.9243 | 0.2071 | 0.2948 | 0.8761 | 0.9152 | 0.2329 | 0.2171 | 0.8319 |
| BLSTM | 0.9341 | 0.1723 | 0.3942 | 0.9282 | 0.9187 | 0.2297 | 0.2719 | 0.8493 |
| CNN+LSTM | 0.9392 | 0.1684 | 0.4707 | 0.9293 | **0.9214** | **0.2129** | 0.3308 | **0.8693** |
| CNN+BLSTM | 0.9356 | 0.1742 | 0.4295 | 0.9271 | 0.9189 | 0.2196 | 0.2893 | 0.8571 |
| SVM WD | **0.9999** | **0.0030** | **0.9995** | **0.9999** | 0.9142 | 2.9621 | 0.3397 | 0.8145 |
| SVM RBF | 0.9453 | 1.8867 | 0.5650 | 0.9999 | 0.9089 | 3.1453 | 0.0112 | 0.8053 |
| Casc. SVM WD | 0.9997 | 0.0082 | 0.9995 | 0.9999 | 0.9123 | 3.0257 | **0.3463** | 0.7981 |
| Casc. SVM RBF | 0.7733 | 7.8276 | 0.3539 | 0.7908 | 0.7124 | 9.9324 | 0.3153 | 0.7693 |

**Notes.**
The best values are highlighted in bold.
Acc, Accuracy; BC, Binary Cross-Entropy; F1, F1-metric; AUC ROC, Area Under the Receiver Operating Characteristic Curve.

**Table 4** Results using 50% of the data.

| Classifier | Train | | | | Test | | | |
|---|---|---|---|---|---|---|---|---|
| | Acc | BC | F1 | AUC ROC | Acc | BC | F1 | AUC ROC |
| CNN | 0.9280 | 0.1947 | 0.3704 | 0.8954 | 0.9189 | 0.2823 | 0.2929 | 0.8645 |
| LSTM | 0.9444 | 0.1565 | 0.5086 | 0.9331 | 0.9294 | 0.1996 | 0.3967 | 0.8824 |
| BLSTM | 0.9488 | 0.1407 | 0.5596 | 0.9482 | **0.9341** | 0.1850 | **0.4489** | 0.9028 |
| CNN+LSTM | 0.9318 | 0.1834 | 0.4076 | 0.9105 | 0.9266 | 0.1976 | 0.3523 | 0.8918 |
| CNN+BLSTM | 0.9416 | 0.1587 | 0.5298 | 0.9340 | 0.9311 | **0.1844** | 0.4418 | **0.9074** |
| SVM WD | **0.9996** | **0.0105** | 0.9983 | **0.9998** | 0.9201 | 2.7569 | 0.4118 | 0.8411 |
| SVM RBF | 0.9521 | 1.6519 | 0.6457 | 0.9997 | 0.9105 | 3.0893 | 0.0683 | 0.8434 |
| Cas. SVM WD | 0.9991 | 0.0285 | **0.9986** | **0.9998** | 0.9118 | 3.0461 | 0.4056 | 0.8142 |
| Cas. SVM RBF | 0.6296 | 12.7913 | 0.2821 | 0.7848 | 0.5343 | 16.0832 | 0.2393 | 0.7702 |

**Notes.**
The best values are highlighted in bold.
Acc, Accuracy; BC, Binary Cross-Entropy; F1, F1-metric; AUC ROC, Area Under the Receiver Operating Characteristic Curve.

Once SVM has been discarded as an adequate model for this problem, our purpose is to analyze which of the five different architectures based on DNNs gets the best results, now using the entire dataset. In Fig. 5 and Table 5, we can see how the CNN model is unable to keep a similar performance to the rest of the methods. Apparently, the model is too simple to detect complex patterns in the sequence. On the other hand, the addition of LSTMs makes the models achieve better results, even higher than the algorithms including a convolutional step in their architecture. This CNN addition makes the model a bit faster by reducing the sequence length given to the LSTM, thus improving GPU parallelization, but its results are slightly inferior. Something that actually helps the model is the bidirectional structure of the LSTM, being the best model for every metric analyzed. This makes its

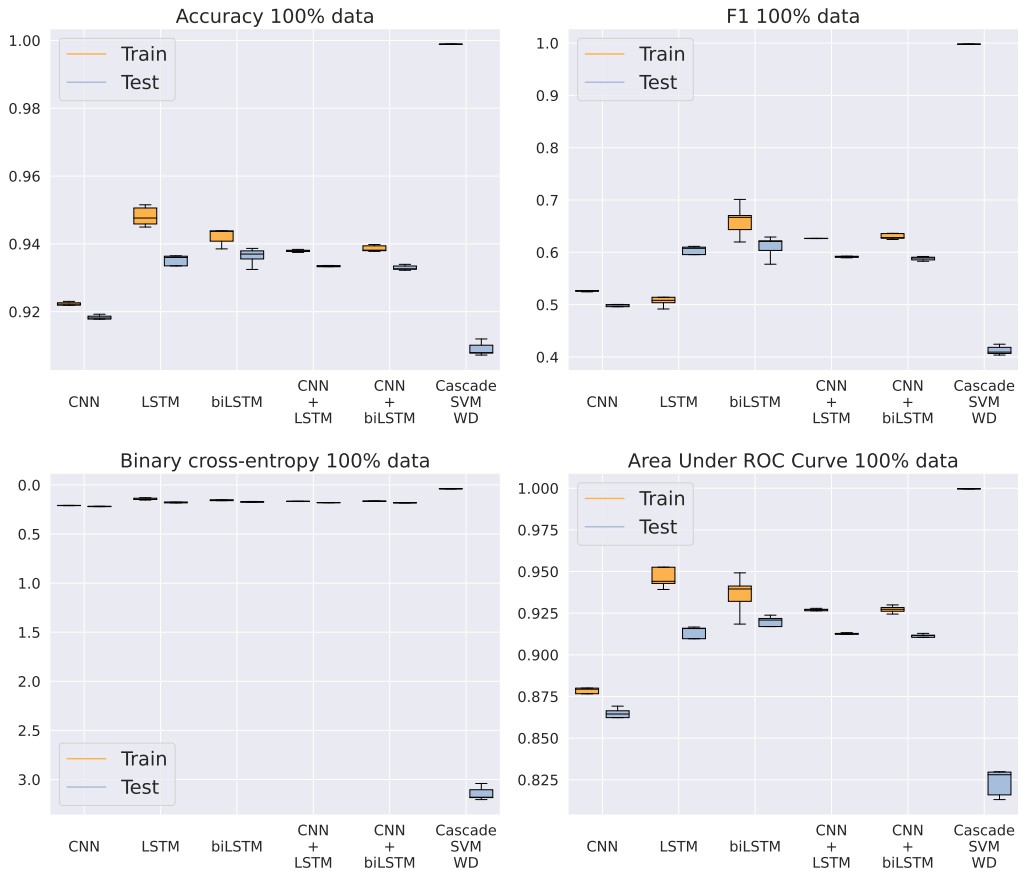

**Figure 5** **Results using 100% of the data.**

training twice as slow as the standard LSTM, from an average training time of 6.69 to 11.4 h using an NVIDIA Titan Xp GPU, that is, a 70% increase in time. Because of the significant difference in results between the RBF kernel and the rest of the methods showed in Table 5, the RBF SVM is not included in Fig. 5 to provide a clearer visualization of the other algorithms.

## Generalization performance

In addition to the performance analysis of the models using the human genome dataset, to test if the obtained models could be used on other datasets without retraining, we created a new dataset from the mouse genome data in Ensembl. The process followed to generate this dataset was the same as already described in Section 2.1, and the dataset is also available from https://github.com/JoseBarbero/EnsemblTSSPrediction.

The results of this last experiment are shown in Table 6. The performance degrades noticeably for all models, something that should not be a surprise, as we are using them on sequences of a different species than the one used to train them. But within this worse performance, it is interesting to see how the SVM models achieve slightly better results than the rest of the models in accuracy and binary cross-entropy, specifically the string

**Table 5  Results using 100% of the data.**

| Classifier | Train | | | | Test | | | |
|---|---|---|---|---|---|---|---|---|
| | **Acc** | **BC** | **F1** | **AUC ROC** | **Acc** | **BC** | **F1** | **AUC ROC** |
| CNN | 0.9210 | 0.2085 | 0.5273 | 0.8818 | 0.9176 | 0.2199 | 0.4968 | 0.8682 |
| LSTM | 0.9470 | 0.1454 | 0.5105 | 0.9438 | 0.9354 | 0.1776 | 0.5771 | 0.9144 |
| BLSTM | **0.9474** | **0.1434** | **0.6602** | **0.9454** | **0.9383** | **0.1697** | **0.6105** | **0.9216** |
| CNN+LSTM | 0.9376 | 0.1674 | 0.6256 | 0.9264 | 0.9329 | 0.1806 | 0.5913 | 0.9121 |
| CNN+BLSTM | 0.9382 | 0.1665 | 0.6303 | 0.9269 | 0.9325 | 0.1817 | 0.5880 | 0.9106 |
| Cas. SVM WD | 0.9988 | 0.0405 | 0.9978 | 0.9996 | 0.9090 | 3.1433 | 0.4124 | 0.8232 |
| Cas. SVM RBF | 0.6109 | 13.8642 | 0.2693 | 0.7833 | 0.5044 | 17.1162 | 0.2386 | 0.7737 |

**Notes.**
The best values are highlighted in bold.

**Table 6  Generalization performance on the mouse genome dataset.**

| Classifier | Test | | | |
|---|---|---|---|---|
| | **Acc** | **BC** | **F1** | **AUC ROC** |
| CNN | 0.5477 | 0.5839 | 0.2926 | 0.6780 |
| LSTM | 0.5479 | 0.6910 | **0.4772** | **0.8288** |
| BLSTM | 0.5477 | 0.5994 | 0.3104 | 0.6984 |
| CNN+LSTM | 0.5479 | 0.6026 | 0.3118 | 0.7143 |
| CNN+BLSTM | 0.5478 | 0.6095 | 0.3226 | 0.7199 |
| Cas. SVM WD | **0.5487** | **0.4656** | 0.2109 | 0.5939 |
| Cas. SVM RBF | 0.5477 | 0.5676 | 0.2784 | 0.6592 |

**Notes.**
The best values are highlighted in bold.

kernel. Although, if we focus on the metrics best suited for unbalanced data, we can see how the models that stand out are actually those using deep learning. In particular, it is surprising how, according to the AUC-ROC metric, the LSTM has a generalization far above the rest of the methods. The LSTM model has the simplest architecture of all the deep neural networks evaluated, which could explain a lower overfit to the training dataset leading to a better generalization ability, extending even to sequences of other species.

## DISCUSSION

Even though the cascade implementation makes a major difference in SVM training times, the poor performance of SVMs on large datasets is well known. In addition to parallelization and the cascade optimization, other methods have been studied trying to improve SVMs performance on large-scale data (*Menon, 2009*; *Saha et al., 2021*). However, they are often not well suited to string kernels and make the model much more complex to implement. Their high computational cost on large datasets makes SVMs hard to apply, existing deep neural networks, being more efficient, and simpler to implement, although

the interpretability and explainability of SVMs are vastly superior, which still makes them the best alternative for certain problems.

On the contrary, DNNs are able to take advantage of large amounts of data, some of them in a more efficient way than others. Specifically, methods using LSTMs can be much slower, particularly the bidirectional ones. That inefficiency is explained by the complexities of parallelizing this kind of models using a GPU, which can be easily solved in models like CNNs.

In terms of performance, bidirectional LSTM has the best results in the human genome dataset , closely followed by the regular LSTM model. From these results, we can derive that the improvement may come from the ability of bidirectional LSTM to consider relevant information not only from the 5′end to the 3′end of the sequence, but also from the 3′to the 5′end, allowing one to find patterns from the 3′end that influence the sequence upstream. This makes us wonder if a slight improvement in the results is worth almost doubling the training time. Given the similarity in performance between BLSTM and LSTM, we performed a Student's $t$-test and concluded that the difference between them is not statistically significant (for $\alpha = 0.05$). Moreover, LSTM got much better results than BLSTM when the models obtained from training with the human genome were used on a different species genome, that of the mouse. Taking these aspects into account, we conclude that the LSTM model would be a better choice in this context.

It is also worth mentioning that more complex architectures, such as combined LSTMs and CNNs, used in similar problems (*Wei et al., 2021*; *Zhang et al., 2017*; *Zuallaert et al., 2018*) did not get the same results as a simpler model like the LSTM. This is probably caused by the loss of information when applying the convolutional and max pooling filters, which indicates a high sensitivity of the model to changes on single bases.

## CONCLUSIONS

This article has evaluated the performance of SVMs and various DNN models for the problem of identifying the transcription start site using our own custom-built datasets based on open data available in Ensembl (*Howe et al., 2020*). These datasets and the instructions to generate them for any other species are publicly available on GitHub and can be used in future papers in this field. Due to their sizes, they are suitable to be use in combination with deep learning methods. These datasets are very simple to download and use, or even generate from scratch, if any researcher wants to alter any step of the process. We have also studied the correct ratio of positive and negative instances in these datasets and concluded that a bigger ratio of negative semi-synthetically generated instances can provide valuable information to the model.

Although SVMs are the most established method in TSS identification, we found that they are poorly adapted to this problem compared to deep learning methods, even those based on string kernels. Meanwhile, DNNs can be much more efficient working with large datasets. We also found that simpler deep learning architectures can work better for this problem than more complicated ones.

We can finally conclude that the SVMs, despite their interpretability and explainability, could not keep up with the DNNs. However, the most complex models that combine

LSTMs and CNNs are not always the best solution. What is clear is that there are models best suited for sequences like LSTMs or SVMs with string kernels, and models best suited for large amounts of data, like those based on DNNs. If we get the best from both perspectives, we can comprehend how LSTMs are the best alternatives in these kinds of problems.

Lastly, we suggest some future research lines for this problem. The first could be to test different encoding techniques, as proposed in *Wu et al. (2021)*. Another interesting work line is the adaptation of popular recent NLP models like BERT, as suggested in *Ji et al. (2021)* to this specific problem. Although transformers have been shown to be effective in this type of tasks, they can be overcome by simpler models like gated multilayer perceptrons, as studied in *Liu et al. (2021)*. One last interesting research line would be to apply these mentioned models to the transcription start site problem, having the opportunity to study the relationship between complexity and efficiency of the methods in this problem.

### Funding

This work has been supported by the Junta de Castilla y León under project BU055P20 (JCyL/FEDER, UE), by the Ministry of Science and Innovation under project PID2020-119894GB-I00, co-financed through European Union FEDER funds and by Fundación Bancaria Caixa under project 2020/00062/001. José A. Barbero-Aparicio is founded through a pre-doctoral grant by the University of Burgos and Alicia Olivares-Gil is founded by the predoctoral grant from the Department of Education of Junta de Castilla y León (VA) (ORDEN EDU/875/2021) (Spain). NVIDIA Corporation donated the TITAN Xp GPUs used in this research. The funders had no role in study design, data collection and analysis, decision to publish, or preparation of the manuscript.

### Grant Disclosures

The following grant information was disclosed by the authors:
The Junta de Castilla y León under project BU055P20 (JCyL/FEDER, UE), by the Ministry of Science and Innovation: PID2020-119894GB-I00.
European Union FEDER funds.
Fundación Bancaria Caixa: 2020/00062/001.
The University of Burgos.
The Department of Education of Junta de Castilla y León (VA) (ORDEN EDU/875/2021) (Spain).
NVIDIA Corporation donated the TITAN Xp GPUs.

### Competing Interests

The authors declare there are no competing interests.

### Author Contributions

- José A. Barbero-Aparicio conceived and designed the experiments, performed the experiments, analyzed the data, performed the computation work, prepared figures and/or tables, authored or reviewed drafts of the article, and approved the final draft.

- Alicia Olivares-Gil analyzed the data, prepared figures and/or tables, authored or reviewed drafts of the article, and approved the final draft.
- José F. Díez-Pastor conceived and designed the experiments, analyzed the data, authored or reviewed drafts of the article, and approved the final draft.
- César García-Osorio conceived and designed the experiments, analyzed the data, authored or reviewed drafts of the article, and approved the final draft.

## Data Availability

The human genome dataset is available at Zenodo:

José A. Barbero-Aparicio, Alicia Olivares-Gil, José F. Díez-Pastor, & César García-Osorio. (2022). Ensembl TSS dataset ofr GRCh38 [Data set]. Zenodo. https://doi.org/10.5281/zenodo.7147597

The mouse genome dataset is available at Zenodo:

José A. Barbero-Aparicio, Alicia Olivares-Gil, José F. Díez-Pastor, & César García-Osorio. (2023). Ensembl TSS dataset for mouse genome. https://doi.org/10.5281/zenodo.7679000

The code is available at GitHub and Zenodo: https://github.com/JoseBarbero/EnsemblTSSPrediction

José Antonio Barbero Aparicio, & cgosorio. (2022). JoseBarbero/EnsemblTSSPrediction: EnsemblTSSPrediction (1.0). Zenodo. https://doi.org/10.5281/zenodo.7224671

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
