# Peer review of "Deep learning and support vector machines for transcription start site identification"

_PeerJ Computer Science, doi:10.7717/peerj-cs.1340_

## Round 0.1 · original submission · Major Revisions

Reviewer 1 asked for new experiments. In my opinion, this makes sense, as it will improve the quality of your proposal. Please address the comments and take their opinions into account.

Reviewer 2 performed a great review, suggesting some key references. Please address the comments and take their opinions into account.

Reviewer 1 ·

Basic reporting

The work titled “Deep learning and support vector machines for transcription start site identification” by Barbero-Aparicio et al. compares the performance of support vector machines and long short-term memory neural networks in the context of transcription start site identification. The text is well-written and easy to read. The work is supported by a sufficient number of references. The structure of the article is standard, the figures are readable and the dataset is shared. In fact, a major highlight of the work is the curation of the used dataset, which is made freely available for the future development and benchmarking of related methodologies.

Experimental design

- For a fair comparison, I suggest the authors explore the performance of SVMs using one-hot encodings as well (with common kernels, such as the RBF).
- When comparing ANN architectures, it is important to remember that the search space is essentially infinite. Hence, it would be preferable to include the criteria used to choose the compared architecture. That is, was the number of trainable parameters fixed among the compared models? Was each architecture optimized on the validation dataset?
- How were the models trained? What batch size, number of epochs, optimizer, and learning rate were used?
- Was early stopping used? Based on what criteria was the training stopped?
- How were the box plots obtained in figures 4 and 5? Were the models retrained with different seeds?
- The results obtained using LSTM and biLSTM should be compared using statistical testing to verify whether the obtained difference is in fact statistically significant.

Validity of the findings

- Could the trained models be used on other datasets without retraining?
- Could the trained models be used in active learning scenarios on datasets where ground truth values are not available?

Reviewer 2 ·

Basic reporting

The paper is well-written, the results are supported with numerical examples, and the references are up-to-date.

Experimental design

The experimental design is in line with other similar work in bioinformatics, but the authors should explore other methods to improve the performance of support vector machines.

Validity of the findings

Conclusions are well-stated and linked to the original research question.

Additional comments

The paper is devoted to comparing the performance between support vector machines and deep neural networks in transcription start site predictions to gene identification. The article is well-written, and the results are supported with numerical examples. However, I have a few essential comments concerning evidencing the results and presentation of the work:

1. It is well-known problem that the SVM algorithm is not suitable for large data sets. The authors mentioned that there are many methods to improve SMVs performance, but they only developed a parallelized algorithm. The authors could use ensemble learning. With ensemble learning the generalization error converges as the number of members increases, guaranteeing that overfitting will not become a problem [1]. Also, it is possible to build a method based on a feature subspace-based ensemble classifier where the large dataset is divided into subsets that are given to a single classifier. Then, an aggregate decision is made by the ensemble voting classifier. There is some research [2] that uses distributed SVMs for solving the problems of sequential SVMs related to loss of classification performance and the high computational cost of SVMs.

2. Although in imbalanced classification domains, the area under the ROC curve (AUC) is a useful metric for classifying performance, using only the ROC curve to select an optimal classifier is not enough, because when AUC has reached a high score, the classification performance may not be as good as AUC value reflects. Authors also use binary cross-entropy as a loss function, but in this type of problem is better to use F-score as an evaluation metric of the models.

3. Common practices for class imbalance problems are balancing classes by using undersampling on majority class instances or oversampling on minority class instances. For evaluation of the ratio in the dataset generation, the authors use undersampling on the majority class, but they do not try to make oversampling on the minority class. Some methods, such as SMOTE [3], keep similar characteristics to generate data perturbed near instances of the minority class. The authors should not discard these types of methods and they should test with their dataset.

4. Please check grammar and other typos. Page 2, line 65: (dduplex…). Page 2, line 77: (His – Their). Page 2, line 90: it is needed a blank space between “perceptrons” and “(Mahdi…). In equation (3), a final parenthesis is needed. Page 12, lines 378-379: “for using” instead “for use”.

[1] Breiman, L. Some Infinity Theory for Predictor Ensembles.

[2] Singh, D., Roy, D., & Mohan, C. K. (2016). DiP-SVM: distribution preserving kernel support vector machine for big data. IEEE Transactions on Big Data, 3(1), 79-90.

[3] Chawla, N. V., Bowyer, K. W., Hall, L. O., & Kegelmeyer, W. P. (2002). SMOTE: synthetic minority over-sampling technique. Journal of artificial intelligence research, 16, 321-357.

---

## Round 0.2 · accepted · Accept

Your manuscript is ready for publication.

Reviewer 2 ·

Basic reporting

no comment

Experimental design

no comment

Validity of the findings

no comment